# Cadence Detection in Road Cycling Using Saddle Tube Motion and Machine Learning

**DOI:** 10.3390/s22166140

**Published:** 2022-08-17

**Authors:** Bernhard Hollaus, Jasper C. Volmer, Thomas Fleischmann

**Affiliations:** 1Department of Medical, Health & Sports Engineering, Management Center Innsbruck, 6020 Innsbruck, Austria; 2Department of Mechatronics, Management Center Innsbruck, 6020 Innsbruck, Austria

**Keywords:** sensor platform, cadence, wearable, machine learning, convolutional neural network, road cycling

## Abstract

Most commercial cadence-measurement systems in road cycling are strictly limited in their function to the measurement of cadence. Other relevant signals, such as roll angle, inclination or a round kick evaluation, cannot be measured with them. This work proposes an alternative cadence-measurement system with less of the mentioned restrictions, without the need for distinct cadence-measurement apparatus attached to the pedal and shaft of the road bicycle. The proposed design applies an inertial measurement unit (IMU) to the seating pole of the bike. In an experiment, the motion data were gathered. A total of four different road cyclists participated in this study to collect different datasets for neural network training and evaluation. In total, over 10 h of road cycling data were recorded and used to train the neural network. The network’s aim was to detect each revolution of the crank within the data. The evaluation of the data has shown that using pure accelerometer data from all three axes led to the best result in combination with the proposed network architecture. A working proof of concept was achieved with an accuracy of approximately 95% on test data. As the proof of concept can also be seen as a new method for measuring cadence, the method was compared with the ground truth. Comparing the ground truth and the predicted cadence, it can be stated that for the relevant range of 50 rpm and above, the prediction over-predicts the cadence with approximately 0.9 rpm with a standard deviation of 2.05 rpm. The results indicate that the proposed design is fully functioning and can be seen as an alternative method to detect the cadence of a road cyclist.

## 1. Introduction

In road cycling, cadence is a metric of high interest [1,2]. Over recent decades, this metric has been studied widely to relate it to cycling performance and efficiency [3,4,5,6]. The state-of-the-art approach [7] in most studies and in commercial cadence-measurement systems uses a Hall-effect sensor [8,9] in combination with a permanent magnet. The Hall-effect sensor is mounted on the bike frame and produces a voltage signal that is proportional to the strength of the magnetic field it is in. A permanent magnet is fixed on the crank or the pedal in such a way that during a whole revolution of the crank, the magnet triggers the sensor only once. This way, the revolutions of the crank can be counted and hence the cadence calculated.

Although this setup is simple and widely used, a drawback exists. This approach is strictly limited to measuring cadence. Other relevant variables such as speed, crank angle, roll and pitch angle, power and torque output or round kick evaluation cannot be measured, though all of them affect the cadence. More advanced measurement systems can measure the power and torque output [10,11,12,13] with strain gauges in addition to cadence. The use of devices based on this method comes with high costs in comparison to the Hall-effect sensor system. Alternatively, cadence can be measured with inertial measurement units. These so-called IMUs detect linear acceleration in multiple directions using accelerometers. In addition, they employ gyroscopes to measure rotational acceleration. For an overview of IMU technologies and applications, see [14]. Companies such as Wahoo or Saris provide IMU-based speed and cadence-measurement systems. These systems also serve as cadence sensors in the studies of Matyja et al. and Gallagher et al. [15,16]. The positioning of the sensors on the bike is strictly limited to the crank arm or the wheel hub for the given products. This position, however, places some restrictions on the sensor that is used. The positioning of a smartwatch or a smartphone on the crank arm, which would make the extra sensor obsolete, is not reasonable. For that reason, a new position of the IMU on the bike is of interest. The new position should allow cadence detection and enable the possibility of using a smart device.

In parallel, cycling has become increasingly digital over recent decades. Not only has training become more digital with the increase in popularity of direct-drive trainers and platforms such as Zwift, but also performance measurements have become more digital, and with it, more advanced. For example, the introduction of wearables to a broad group of athletes have allowed performance metrics, such as VO2max, heart rate, SpO2 and many more, to become widely accessible [17,18,19]. Another device that has aided the digitalisation of cycling is the IMU. It has been used in a variety of studies to assess for example cycling performance. Evans et al. worked on an efficient saddle position for triathlon using an IMU-based system [20]. Marin-Perianu et al. and Maio et al. both worked on estimating cyclists’ lower limb positions using IMUs [21,22]. Several studies have measured surface roughness while cycling with an IMU-based system [23,24,25]. Most notably, Ref. [23] mentions that the pedalling signal is filtered from the data to analyse the vibrations caused by the road using independent component analysis (ICA), yet it is not actively analysed. To the authors’ best knowledge, current literature does not show how to measure the pedalling signal or cadence using an IMU in positions other than the pedals, crank arm or wheel hub.

As cadence is a measure of high interest and IMUs are presently ubiquitous in sports, this work presents a working prototype and a proof of concept of a cadence-measurement system for road cycling based on an inertial measurement unit, where the IMU is mounted on a less exposed position on the bike. The goal of this work is therefore to show that an IMU mounted in such a position is still capable of measuring cadence. It is emphasized that, to our best knowledge, this has neither been attempted nor successfully proven before. To determine the cadence from the IMU signal, a machine-learning approach is taken, hence the cadence-measurement method is called IMU-ML method in this paper.

The rest of this paper is structured as follows: first, the method for collecting data is shown. Then, the analysis of the data using neural networks is explained. Finally, the performance of the neural network is analysed, and the IMU-ML method is compared to the ground truth.

## 2. Methods

To provide a proof of concept for the IMU-based cadence-measurement system of road bikes, it is necessary to gather data from both the IMU on the saddle tube and the state-of-the-art approach with a Hall-effect sensor and a permanent magnet. First, a sensor platform was chosen, configured and programmed to be used as data logger during the experimental bike rides. After the data were gathered using this platform, an algorithm had to be developed that takes the IMU data as input and calculates the corresponding cadence as output, according to the ground truth of the Hall-effect sensor. To make the methods section concise, we split it into a hardware/firmware experiment and software.

### 2.1. Hardware/Firmware

At the beginning of the data-gathering stage, a data logger was developed that records the IMU data and the Hall-effect sensor data in a synchronized fashion with a sufficiently high sampling rate. At the same time, the data logger had to be lightweight, programmable and small. In other sports such as tennis and American football, the requirements are similar. Based on the paper by [26,27], the sensor platform for this work has been chosen. Nevertheless, adaption to hardware and firmware had to be made for the specific needs in the experiment.

The sensor platform in this work is based on the SensorTile [28], which contains the IMU. Figure 1 shows the sensor platform with marked components and the connection to the Hall-effect sensor [29]. The SensorTile module is soldered to the SensorTile Cradle board, which provides power via the battery and the micro-SD card slot to the microcontroller system. To measure the cadence with the standard state-of-the-art approach, the signal of the Hall-effect sensor has been processed as well. As it is an analog signal, it is connected to the internal analog-to-digital converter (ADC) of the microcontroller. The sensors in Table 1 are connected to the MCU via the SensorTile module PCB.

The setup and mounting of the hardware was undertaken according to Figure 2 for all cyclists and bikes during the experiment.

The measured data were stored in a *.csv file on the attached micro-SD card. The given experimental setup enabled the recording of the following data

Time since startup in ms (T),Accelerometer X axis in mg (AccX),Accelerometer Y axis in mg (AccY),Accelerometer Z axis in mg (AccZ),Gyroscope X axis in $m^{\circ }\;s^{-1}$ (GyroX),Gyroscope Y axis in $m^{\circ }\;s^{-1}$ (GyroY),Gyroscope Z axis in $m^{\circ }\;s^{-1}$ (GyroZ),Magnetometer X axis in mG (MagX),Magnetometer Y axis in mG (MagY),Magnetometer Z axis in mG (MagZ),Air pressure in mbar (P),Hall Effect as raw ADC 12-bit value (Hall)

Based on the hardware setup, the firmware had to fit the needs of the experiment. The firmware on the microcontroller is based on the STSW-STLKT01-DataLog example [30] provided by STMicroelectronics. It is based on FreeRTOS (free real-time operating system) and works with two threads—one used to collect and sample the data and the other one to store it on the micro-SD card. The thread responsible for data sampling is set to a higher priority. The scheduling between the two threads is undertaken as in the example project of STMicroelectronics.

Based on the example, further adaption of the firmware was necessary. The initialization routine was rewritten so the timer of the data logger was able to sample with up to 1 kHz. Although the authors expected the cadence in the range of several Hz maximum, a high sampling rate presents possibilities at the data-processing stage. The Hall-effect sensor used the internal 12-bit ADC, which had to be configured in the initialization routine. In addition, the IMU and the pressure sensor required a specific configuration, which was covered in the initialization routine. During this routine, the sensors were set as listed in Table 2.

With the adaptions in the initialization routine done, the recording and saving of each sample on the SD card was slightly altered as well. All sensors were recorded and saved with a comprehensive time stamp in a *.csv file. Since threading was used, the time between samples was not constant, but never exceeded 20 ms. The aforementioned pressure sensor was initialized, recorded, and saved, but, due to non-reliable measurement results, no longer considered for the final dataset.

From a firmware sequence perspective, the initialization routine is triggered directly after the data logger is switched on. After initialization, the data logger automatically starts collecting data if a micro-SD card is available. The recording stops automatically after a defined time span, which was chosen to be one hour for the experiment.

### 2.2. Experiment

With the data logger ready, an experiment was carried out to gather data. All participants were instructed to use a trigger routine at the beginning of each ride. To ensure that the final dataset consists of the activity of road cycling only, the start of the bike ride had to be marked within the gathered data of the experiment. To do so, the magnet excites the Hall-effect sensor for at least five seconds, just by putting the crank arm in the respective position. This enables a defined trimming of the data as a saturated signal from the Hall-effect sensor for at least five seconds does not occur during a normal ride with a road bicycle.

As the goal was to provide a proof of concept, the group of participants, chosen to do experiments, was small. In total, four road cyclists comprising males and females with an age ranging from 23 to 57 made a total of 13 rides leading to 642 min of data. All participants were instructed to ride any route they want. No restrictions were imposed on the routes ridden. However, in practice, virtually all rides were performed in the vicinity of the city of Innsbruck. Therefore, participants encountered relatively good-quality asphalt roads and significant height differences leading to a varied cadence dataset. Before the experiment took place, it was approved by the ethics committee of the MCI. All participants signed a consent form.

### 2.3. Data Preparation

After the data-gathering stage, an algorithm had to be developed so the cadence could be derived based on the IMU data. The authors chose a supervised machine-learning approach as found in many other sport-related studies such as [26,27,31], due to their good performance. To this end, the individual pedal strikes that the Hall-effect sensor detects were used as the ground truth. As the Hall-effect sensor and the IMU sensor are processed on the same hardware, the timesteps are synchronized. This, in turn, provides labels (pedal strikes) for the IMU data-streams at every given time, and avoids the need for manual labelling.

As a first step, the raw data had to be processed to become a suitable dataset for the network. This was undertaken with a Python script programmed and run inside Jupiter Notebook [32]. The script took care of importing the raw data, trimming, reshaping and normalizing it. Data were also checked for whether they suited the requirements for a neural network development. An analysis of the Hall-effect sensor signal showed that it rarely is greater than 0. Indeed, the normalized Hall-effect sensor only outputs 1 when the pedal passes the sensor. This leads to a mean value of the signal over all rides of close to 0. In addition, the data also showed a variety of timesteps between samples. Two measures have been taken to avoid further problems with the gathered data.

To overcome the timestep issue, all sensor signals have been resampled to 50 Hz by calculating the mean value of all given values for the respective signals within 20 ms windows. This way, a constant sampling rate was achieved based on signals that had no constant sampling time, but were recorded so at least one value was recorded in each 20 ms window.

To avoid a highly unbalanced dataset and reduce the noise in the signal, the Hall-effect sensor signal was adjusted. The Hall-effect sensor signal was processed with the peakfinder function from the Python library scipy [33] to extract the peaks created by the permanent magnet on the crank. After the peaks were identified, a convolution of a 260 ms wide impulse was undertaken, so each original peak in the pedal stroke signal became a 260 ms wide impulse with an amplitude of 1. Therefore, the resulting signal is either 0 or 1. After this adaption, the mean value of the pedal stroke signal was approximately 0.22, so the optimizer is forced to consider the impulses in its training to achieve accuracies for the prediction better than 78%. If the signal’s value is considered to be a label, it can be said that 22% of all labels are of Class 1, and 78% are of Class 0.

For training, validation, and testing the network, the data were split into training (70%), validation (15%), and test (15%) datasets.

### 2.4. Network Architecture

As the data were processed and split to suit the training process of a neural network, a network was designed, developed, validated and tested in a second step. A common approach for time series classification is the so-called convolutional neural network (CNN, [34]) with a long short-term memory (LSTM, [35]) approach [36,37,38]. The authors chose to use this approach to provide proof of concept. The purpose of this network is to generate a pedal stroke pattern comparable to the values provided by the preprocessed Hall-effect sensor signal. Therefore, the IMU data are the network’s input, and the pedal stroke signal represents the output.

Figure 3 shows the final structure of the used network, containing two convolutional layers followed by an LSTM layer. Primarily, convolutional layers are used as feature extraction layers. The development of the features over time is interpreted by the LSTM. Dense layers are used to combine either input signals, to create a wider spectrum of possibilities for the convolutional layers, or to summarize the outputs of the LSTM to one output. As can be seen in Figure 3, only three input signals (acceleration in X, Y, and Z) were considered for the final network from the 11 available. The question marks in Figure 3 symbolise the unknown batch size at the creation of the neural network. This means that the network can accept any number of samples in a given dataset. The rectified linear unit (ReLU) [39] was the commonly used activation function for all layers, except for the last dense layer, which used a sigmoid activation function. This structure led to 110,209 trainable parameters. During the training stage, the Adam optimizer [40] was used with mean squared error (MSE) as a loss function, which was computed as
(1)$$\mathrm{MSE}=\frac{1}{n}\sum_{i=1}^{n}{\left(p_{i}-\hat{p_{i}}\right)}^{2},$$
with *n* the number of data points, $p_{i}$ the processed pedal stroke signal from the Hall-effect sensor (ground truth), and $\hat{p_{i}}$ the predicted pedal stroke signal from the network. In addition to the MSE, a binary accuracy was computed as the ratio between the number of correct predictions to the total number of predictions. To this end, the continuous prediction produced by the network is mapped to a binary output using a threshold value of 0.5, i.e., any value above this threshold is mapped to 1 and any value below is mapped to 0.

Neural networks are well known to show poor generalization behaviour when trained incorrectly. This effect is termed overfitting, and must be prevented. Several methods exist; see [41] for an overview. We opt for an approach that terminates the network training before it converges, based on validation performance [42,43]. This so-called ’early stopping’ was implemented via a callback that tracks the minimal validation loss with a patience of one epoch.

Since the goal is to develop a working prototype and a proof of concept of a cadence-measurement system for road cycling based on an inertial measurement unit in a less exposed position on the bike, the last step had to be the comparison of the ground truth with the IMU-ML system. To do so, the cadence is calculated in the same way for both systems using
(2)$$c=num_{p}/60,$$
with *c* as cadence and $num_{p}$ as number of peaks, with a moving window approach.

## 3. Results

To recap, the IMU-ML network described in the previous section is used to predict a pedal stroke signal comparable to that of the Hall-effect sensor based on the IMU data. To show the performance of the IMU-ML, this section compares its performance to the ground truth recorded with the Hall-effect sensor. First, the performance of the neural network is shown in Table 3.

The performance shows that the accuracy is high for both the training validation and test set. Therefore, overfitting was prevented due to the early stopping callback. At the same time, the accuracy does not directly reflect if the IMU-ML method performs well as a cadence-measurement system. Table 3 just shows that the pedal stroke signal can be predicted quite well. Figure 4 gives an example of the pedal stroke signal as predicted. It can be seen that the network output corresponds well with the sensor signal. In addition, the grey dashed line indicates the binary threshold as used in the binary accuracy metric.

The calculation of the cadence using (Equation 2) and the pedal stroke signal needs to be applied on both the ground truth signal and the predicted signal. Figure 5 shows the comparison of the calculated cadence of the ground truth and the IMU-ML method.

The cadence signals that are shown in Figure 5 come from the test dataset. The initial goal of providing a proof of concept is directly supported due to a very close estimation of the ground truth cadence with the IMU-ML method. For better visibility and a better analysis, two Bland–Altmann diagrams show the comparison in more detail. See Figure 6a,b for a more detailed comparison of the two methods.

All the given values in this paragraph are in comparison to the ground truth method. The overall performance of the IMU-ML method for predicting *c* comes with a mean over-prediction of 1.95 rpm and a standard deviation of 5.05 rpm considering the whole range of the dataset. Considering only the most relevant range of 50 rpm and above, the IMU-ML method predicts *c* with approximately 0.9 rpm too high on average with a standard deviation 2.05 rpm.

## 4. Discussion

A major limitation of the experiment, and therefore also for the results, lies in the limited number of participants. With only four participants, the collected dataset is not as comprehensive as it should be for a commercial system. Nevertheless, the goal was to create a proof of concept. The goal was met, as it can be argued that the cadence could be retained from the IMU data with sufficient accuracy. The method can be applied to a wider dataset, resulting in a similar or even better performance.

The participants in the study were not restricted in their cycling routes. The result was that all the participants chose a route on asphalt. Therefore, it must be said that the trained network may perform worse on other surfaces, as the motion of a cyclist and the bike will change.

In the network-development process, the input signals to the network were determined. The question was which signal contained information on cadence and to what extent. Based on an initial assumption that accelerations and spin rates have the biggest correlation to the cadence signal (see Appendix A for a singular value decomposition of the data), an ablation study focused on these signals as single inputs and as combined inputs. Table 4 shows their performance for validation and testing.

It can be argued that spin rate and acceleration correlate well with the cadence signal. The combination of both sensor signals as input to the network also achieves high accuracy, but has the drawback of a much higher number of trainable parameters. For that reason, it was not further developed, as the performance of the accelerometer alone was already sufficiently accurate to support the initial goal to provide a proof of concept.

The structure of the network as given in Figure 3 might not be optimal. Additionally, accuracy as a performance metric on an unbalanced dataset with a binary label can be considered improvable. Additionally, with the concept of hyper-parameter optimization, the network structure could be optimized. Using the F1 score as a performance metric or binary, cross entropy as a loss function might lead to better results. As the goal was to provide the proof of concept for the IMU-ML method and not the development of an optimal neural network, this was not in the scope of the article.

The IMU-ML method performs well in the range of 50 rpm and above, as shown in Figure 6b. As this range is strongly represented in the dataset, it is only a logical result. For a better performance in the whole range, more versatile data must be gathered.

Figure 6a shows some outliers that were mainly overpredicted. These outliers show that the network is not able to process all the IMU data in a reasonable way. The reason for these outliers might lie in the split of the training, validation and test dataset. It is possible that the motion patterns that were mispredicted, have not been part of the training and validation dataset. Therefore, the network does not perform well when predicting the cadence for some of these motion patterns. This is also true for new IMU data of other cyclists compared to the ones that participated in the experiment.

## 5. Conclusions

The present work has described the development of a cadence-measurement system. This system is based on an IMU combined with a CNN-LSTM network. The novelty of this work is two-fold. First, the IMU is mounted behind the seat post as opposed to in an exposed position on the crank. Second, to our best knowledge, deducting cadence from an IMU signal using a CNN-LSTM network has not been shown before. All results considered, it can be stated that the initial goal of the study was met. The IMU-ML method is sufficiently accurate to state the principle functioning of this proposed cadence-measurement tool. For the entire dataset, the method over-predicts the cadence with 1.95 ± 5.05 rpm. However, when only considering the most relevant range of 50 rpm and above, the method is more accurate with an over-prediction of 0.9 ± 2.05 rpm.

The method shown could be extended to use IMUs present in devices such as smartphones or smartwatches. This is supported by the fact that this method does not require the IMU to be mounted on the crank. In addition, an IMU could be integrated into a bike component such as a saddle or a handlebar.

The authors believe that the IMU signal contains far more information than just cadence. It is not unlikely that further cycling metrics could be deducted with a similar approach. Hence, we suggest further research should focus on identifying such performance metrics and show their correlation to cycling efficiency and skill level. At the same time, the IMU-ML method must be further developed, and further research must be done considering the robustness of the cadence prediction. This is possible by including more cyclists on various routes and terrains covering a wide range of scenarios and cadence range.

## Figures and Tables

**Figure 1 sensors-22-06140-f001:**
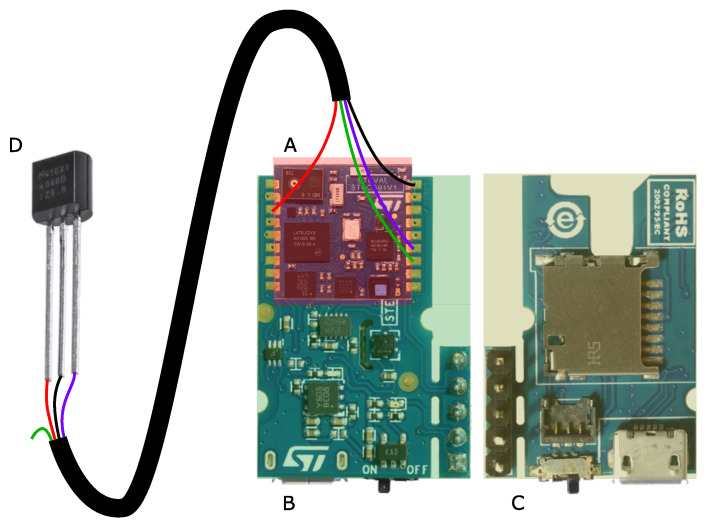
The hardware block diagram with the SensorTile module (**A**) which contains the IMU, the SensorTile Cradle board in top (**B**) and bottom view (**C**) [28] and the Hall-effect sensor (**D**) [29]. The connecting cable between Hall-effect sensor and SensorTile module has the supply voltage in red, the ground in black, the analogous Hall-effect sensor output in violet, and a spare wire in green.

**Figure 2 sensors-22-06140-f002:**
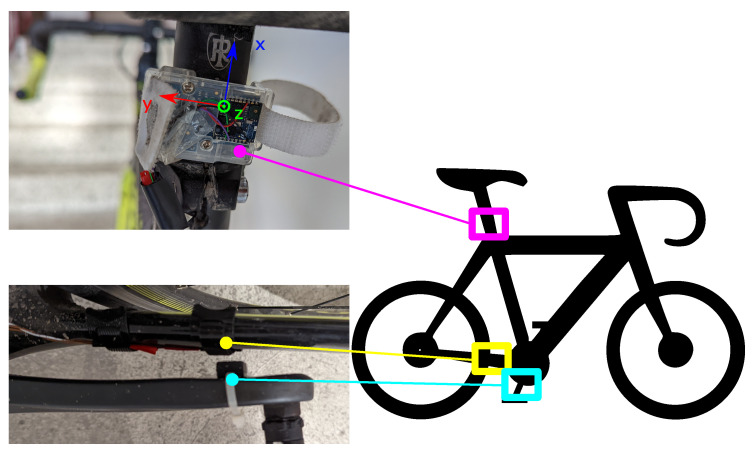
The experimental setup for the measurement of the motion data and the cadence signal. The IMU is fixed to the saddle tube with the given orientation. The Hall-effect sensor is fixed on the frame opposite the permanent magnet on the crank arm.

**Figure 3 sensors-22-06140-f003:**
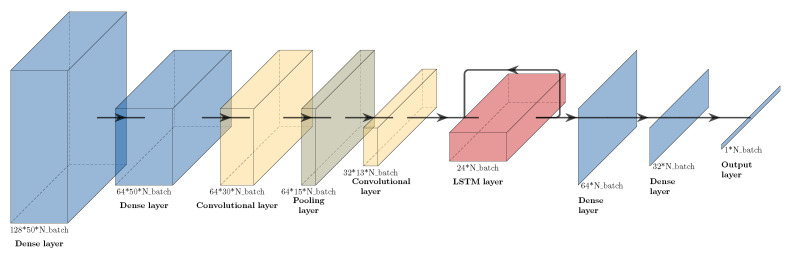
Structure of the neural network model, with layer types and sizes as indicated. $N_{batch}$ symbolises the variable dimension based on the number of batches.

**Figure 4 sensors-22-06140-f004:**
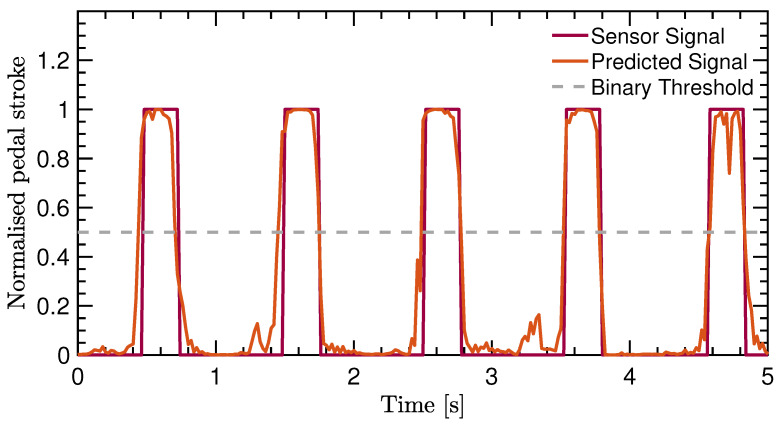
Pedal stroke signal from the Hall-effect sensor after pre-processing versus the output signal predicted by the network. The binary threshold indicates the threshold used to decide if the network output is mapped to 0 or 1.

**Figure 5 sensors-22-06140-f005:**
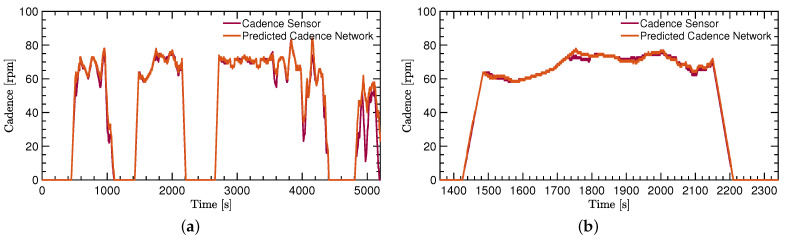
Neural network evaluation: true cadence as measured with the Hall-effect sensor versus predicted cadence from the network. (**a**) Comparison for a large section from the test dataset. (**b**) Comparison for a zoomed section from the test dataset.

**Figure 6 sensors-22-06140-f006:**
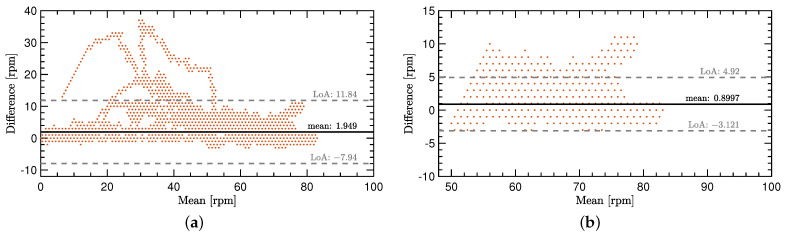
Bland–Altmann diagrams for the comparison of the IMU-ML method with the Hall-effect sensor used as ground truth. (**a**) Comparison for the entire cadence range in the dataset. (**b**) Comparison for 50 rpm and above.

**Table 1 sensors-22-06140-t001:** IMU and sensors connected to the SensorTile module and their main characteristics.

Device	Description	Interfaces	Zero-Level Offset
STM32L476JG	ARM Cortex-M4 32-bit microcontroller	I^2^C, SPI, UART, ADC, etc.	-
LSM6DSM	3D accelerometer (up to ±16 g), 3D gyroscope (up to ±2000${}^{\circ }\;s^{-1}$)	I^2^C, SPI	±40 mg, ±2${}^{\circ }\;s^{-1}$
LSM303AGR	3D accelerometer (not used), 3D magnetometer (up to ±50 G)	I^2^C, SPI	±40 mg, ±60 mG
LPS22HB	pressure sensor (up to 1260 hPa absolute pressure range)	I^2^C, SPI	0.0075 hPa (RMS)
DRV5053VA	analog bipolar Hall-effect sensor (−90 mV/mT sensitivity)	analog	1.02 V (typical)

**Table 2 sensors-22-06140-t002:** Sensor devices and their settings as configured in the initialization routine prior to the start of a measurement.

Device	Sensor	Setting
LSM6DSM	Accelerometer	±2 g
LSM6DSM	LGyroscope	±2000 ${}^{\circ }\;s^{-1}$
LSM303AGR	Magnetometer	±50 Gauss
LPS22HB	Barometer	260 hPa to 1260 hPa absolute pressure range
DRV5053VA	Hall effect	Provides raw 12-bit unsigned ADC values

**Table 3 sensors-22-06140-t003:** Neural network evaluation showing the loss (MSE) and accuracy (Binary) for the final network for the training, validation, and test datasets.

Loss	Accuracy	Val_loss	Val_acc.	Test_loss	Test_acc.
0.0525	0.9291	0.0551	0.9264	0.0367	0.9506

**Table 4 sensors-22-06140-t004:** Neural network evaluation showing the loss (MSE) and accuracy (Binary) for network using various sensors as input for the training, validation and test datasets.

Input Sensor	Validation Loss	Validation Accuracy	Test Loss	Test Accuracy
Accelerometer	0.0551	0.9264	0.0367	0.9506
Gyroscope	0.3805	0.8731	0.3561	0.8859
Accel. and Gyro.	0.2860	0.8902	0.2817	0.8910

## Data Availability

Data is available from the authors.

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
