# Peer review of "Cadence Detection in Road Cycling Using Saddle Tube Motion and Machine Learning"

_sensors, 2022, doi:10.3390/s22166140_

Round 1

Reviewer 1 Report

In this work, cadence measurement system using Inertial measurement unit (IMU) installed in the seating pole of the bike was employed. The work is well organized and a sufficient number of experiments. There are a few suggestions and queries that should be incorporated in the revised manuscript.

1. There is no mathematical formulations and equations that should show the relationship between cadence and other parameters..

2. Validation samples are normally selected from the training samples. Normally, 70% training samples are then divided into training and validation samples. Is there any reference to divide data into training, validation and test samples.

3. How the overfitting is avoided. There should be a detailed explanation for avoiding overfitting.

4. In real situations, what about protections of the sensors in un controlled situations.

5. Figure 5 shows the structure of neural network. How LSTM gates are incorporated in the given diagram.

6. To prevent outliers, feature selection techniques can be used and a very simple PCA analysis can be performed. Please incorporate such a technique in the revised manuscript.

Author Response

Dear reviewer,

thank you for your review. We have incorperated your comments into our revised manuscript, which we have sent to Ms. Ivory Feng. Please find attached the explainations and our responses to your review. We hope to satisfy your expectation. 

BR the authors

Reviewer 2 Report

a). An interesting work which presents a working prototype and a proof of concept of a cadence measurement system for road cyclist based on an IMU.

b). Method evaluation: authors should provide the comparison of IMU-ML method with other methods mentioned in Section 1.

Author Response

(The authors gave the same response as above.)

Reviewer 3 Report

Section 1 must be improved.

Authors should emphasize contribution and novelty, the introduction needs to clarify the motivation, challenges, contribution, objectives, and significance/implication.

You should introduce the problem in more detail so that the reader is immediately clear about the purpose of your study. Specify better the essential elements of the problem. You should add more information in the introductory part, you should add other works that have also addressed the problem.

You must properly introduce your work, specify well what were the goals you set yourself and how you approached the problem.

At the end of the section, add an outline of the rest of the paper, in this way the reader will be introduced to the content of the following sections.

Section 2 must be improved.

Describe in detail the equipment used to make the measurements (IMU and hall effect sensor). Extract this data from the datasheet of the instrumentation manufacturer. To make reading the specifications of the instruments more immediate, you can insert them in a table, listing the instruments used and the specific characteristics for each.

In the experiment subsection you should describe in detail how the participants conducted the experiment. What type of route did they perform, if they simulated different cadences, which ones, if they maintained a specific speed. You should add information about the participants: e.g. age, gender, type of job etc.

Also, I haven't found a section devoted to data labeling. In the title you mention supervised learning, so this means that you have labeled data at your disposal. The success of a machine learning-based method depends on the quality of the data. This quality in the case of supervised learning is essentially linked to the goodness of the labeling. Add this section.

I could not find a detailed description of the evaluation metrics you have adopted. How will you measure your model's performance? This section is essential in order to demonstrate the effectiveness of your methodology. Furthermore, only by adopting adequate metrics will it be possible to compare your results with those obtained by other researchers.

Section 3 must be improved. The results need to be described in more detail. First, summarize the experimental procedure and then describe how you performed the analysis. Then present your performance metrics. In figure 2 you have shown two sensors (IMU and hall effect sensor), where are the comparisons obtained with the two sensors? Compare your results with those in the bibliography, if you do not find other works that have addressed your problem you can refer to CNN and LSTM generally.

Section 4 must be improved. Merge sections 3 and 4 into one and rename it results and Discussions

Section 5 must be improved. This section is too succinct. You need to summarize your initial goals and what results you have achieved. then add your conclusions. Paragraphs are missing where the possible practical applications of the results of this study are reported. What these results can serve the people, it is necessary to insert possible uses of this study that justify their publication. They also lack the possible future goals of this work. Do the authors plan to continue their research on this topic?

24)” hall effect sensor” Introduce adequately the topic. Add references to allow readers to learn more about the topic.

35)” inertial measurement units (IMU)” Introduce adequately the topic. Add references to allow readers to learn more about the topic.

55) Do not use abbreviation such as i.e. I have seen that you often use this abbreviation, so I will not repeat this advice again, it also applies to the other occurrences.

171) Introduce adequately the topic (CNN and LSTM). Add references to allow readers to learn more about the topic.

197)” evaluation - metrics” Introduce adequately the topic.

Author Response

(The authors gave the same response as above.)

Reviewer 4 Report

The proposed manuscript deal with an interesting machine learning approach to cadence detection in road cycling.

 The paper is clear, readable, well organized and written. The abstract and introduction sections are well focused on the paper topic.

Even if the general paper frame is really interesting, the paper is missing crucial information about the hypothesis and methodological approach and doesn’t provide the full info needed to evaluate it.

More in details:

The authors don’t specify information about the bike type and the cyclists and the if this information have or not an influence on the results.

It is not clear if the authors used only road bike and why.

All the rider chosen a route on asphalt limiting the papers result to a specific path type.

I would suggest to improve the dataset:

-       improve the number of riders

-       increase the time of each record

-       use different bike type (road, gravel, MTB with front suspension and/or full suspended, …)

-       Differentiate the route

-       Improve the “RPM” recorded range to make the data more realistic. There are cyclists who reach up to 160 RPM during a ride. 

My personal opinion is that with more data the paper il be more robust and valid from scientific point of view.

In the present form, the paper is eligible for publications on Remote Sensing only after major revision.

Author Response

(The authors gave the same response as above.)

Round 2

Reviewer 3 Report

The authors addressed the reviewer's comments with attention and modified the paper with the suggestions provided. The new version of the paper has improved both in the presentation that is now much more accessible even by a reader not expert in the sector, and in the contents that now appear much more incisive.

Minor revision

- displays the tables in the format expected by the journal

Try to enrich the captions of the figures, the reader should be able to read the figure without the need to retrieve the information in the paper. Try to summarize the essential parts of the Figure and what you want to explain with it.

 - (122-126) Add these information as table

- 242) enter a description of both subplots (a, b) in the caption. I have seen that you often use this format, so I will not repeat this advice again, it also applies to the other occurrences.

Author Response

Dear reviewer,

Thank you again for your excellent comments, we have included the minor revisions by:

  • Updating the table lay-outs to the booktabs format used by the journal
  • Expanding the captions of tables 2, 3 and 4 and figure captions 3, 5a, 5b and 6.
  • Listing the information from line 122-126 in a new table 2.
  • Adding a caption to all subplots.

Best regards,

The authors

Reviewer 4 Report

The paper ready to be published in the present form.

Author Response

Thank you for the reviewing of the paper. We are happy to satisfy your requirements.